# Primary Healthcare Providers’ Views on Periodic COVID-19 Booster Vaccination for Themselves and Their Patients: A 2023 Nationwide Survey in Belgium

**DOI:** 10.3390/vaccines12070740

**Published:** 2024-07-03

**Authors:** Marina Digregorio, Pauline Van Ngoc, Julie Domen, Zsofia Bognar, Els Duysburgh, Greet Hendrickx, Pierre Van Damme, Samuel Coenen, Beatrice Scholtes

**Affiliations:** 1Research Unit of Primary Care and Health, Department of General Medicine, Faculty of Medicine, University of Liège, 4000 Liège, Belgium; 2Department of Family Medicine and Population Health (FAMPOP), Centre for General Practice, University of Antwerp, 2610 Antwerpen, Belgium; 3Department of Epidemiology and Public Health, Sciensano, 1050 Brussels, Belgium; 4ECDC Fellowship Programme, Field Epidemiology Path (EPIET), European Centre for Disease Prevention and Control (ECDC), 171 83 Stockholm, Sweden; 5Centre for the Evaluation of Vaccination Vaccine & Infectious Disease Institute, University of Antwerp, 2610 Antwerpen, Belgium

**Keywords:** COVID-19, COVID-19 booster vaccine, periodic booster dose, primary healthcare providers, general practitioner, Belgium, COVID-19 vaccination, vaccine hesitancy

## Abstract

New COVID-19 strains and waning vaccine effectiveness prompted initiatives for booster vaccination. In Belgium, healthcare providers (HCPs) received a second booster in July 2022, with eligible individuals receiving a third in autumn. Primary HCPs (PHCPs) play a crucial role in healthcare organization and patient communication. This study, conducted in February–March 2023, surveyed 1900 Belgian PHCPs to assess their views on periodic COVID-19 boosters for themselves and their patients. The survey included questions on sociodemographic information, willingness to receive periodic COVID-19 boosters, reasons for acceptance or refusal, confidence in vaccine safety and efficacy, and views on booster recommendations. Overall, 86% of participants were willing to receive periodic COVID-19 boosters, motivated by self-protection, patient well-being, and the uninterrupted delivery of healthcare services. Factors influencing booster refusal included not being a general practitioner (GP) or GP trainee, working in Wallonia or Brussels, and lacking vaccine confidence. Although 243 participants would not take boosters periodically, only 74 would not recommend it. Regarding administration, 59% supported pharmacist involvement in COVID-19 vaccination. Further qualitative analysis of 290 PHCPs’ responses revealed varying recommendations, including specific roles like nurses, organizational structures, and collaborative approaches. This study highlights the need to address vaccine confidence, regional disparities, and PHCP roles in booster implementation.

## 1. Introduction

To achieve community immunity and in view of the emergence of new variants as well as the rapid decline in vaccine efficacy, COVID-19 booster vaccination campaigns have been launched worldwide [1,2,3]. In July 2022, healthcare providers (HCPs) across Belgium were offered a second booster vaccination. The population, including HCPs who had already received a booster earlier in the year, was eligible for another (third) booster during the 2022 autumn campaign, with a minimum 3-month interval between doses [4]. Belgium periodically reviews its distribution strategies for booster vaccines. For the autumn–winter season of 2023–2024, the Belgian National Immunization Technical Advisory Group (NITAG) established a COVID-19 vaccination plan that prioritizes specific groups using mRNA vaccines [5]. The recommended target population for COVID-19 booster vaccinations during the autumn/winter 2023–2024 season includes individuals at an increased risk of death or serious illness, individuals employed in the healthcare sector, and people living with at-risk people [5].

Primary healthcare providers (PHCPs) play a crucial role in efficiently organizing healthcare services, as they are responsible for handling a significant portion of patients. Among PHCPs, general practitioners (GPs) serve as key healthcare information sources [6,7,8]. Furthermore, insights and recommendations provided by healthcare professionals, especially physicians, hold substantial significance in guiding patients’ vaccination choices [9,10]. Therefore, vaccine refusal or hesitancy among PHCPs can lower vaccine uptake by their patients and increase their risk of infection. PHCPs’ beliefs and attitudes also play an important role in primary prevention and health promotion [11,12,13]. Vaccine hesitancy among HCPs is not a new phenomenon, even though it may not be prevalent. Even before the COVID pandemic, the rate of influenza coverage among HCPs in Europe displayed a declining trend [14]. In addition, prior to the COVID-19 vaccine’s distribution, research showed significant reluctance among HCPs, with the percentage of willingness to receive the vaccine varying across various countries such as France, Belgium, and Canada [15,16]. Pal S. and colleagues also noted that there was not widespread willingness among US HCP to consider a potential COVID-19 booster vaccine during the period from February 1, 2021, to March 31, 2021 [17]. Additionally, the rapid development of COVID-19 vaccines utilizing innovative techniques during the COVID-19 pandemic boosted vaccine hesitancy among HCPs [18].

The current study was carried out in February-March 2023 following the second and third COVID-19 booster vaccine campaigns and preceding the implementation of the 2023–2024 vaccination plan. Its objective is to investigate the views of PHCPs regarding a periodic COVID-19 booster vaccine. We determined the percentage of PHCPs who did not want to receive periodic booster vaccines and identified the individual characteristics that might be linked to this unwillingness. Additionally, we explored the main reasons behind their acceptance or rejection of these periodic booster vaccines. We also examined whether PHCPs have specific target groups in mind when considering booster recommendations, assessed how likely they are to recommend it, and understood the reasons behind their reluctance to make such recommendations (if applicable). Additionally, given the decision in Belgium to allow pharmacists to prescribe and administer COVID-19 vaccines in 2023 under specific circumstances, such as direct administration following prescription, we gathered PHCPs’ opinions on this change [19]. Finally, we examined the sources from which PHCPs gather information regarding COVID-19 and vaccines, identifying their most trustful sources for information.

## 2. Materials and Methods

### 2.1. Study Design and Population

The present study is a sub-study of a national study (CHARMING) in which prevalence, incidence, and long-term presence of antibodies against SARS-CoV-2 both after natural infection and after vaccination were assessed among Belgian PHCPs from a convenience sample of 2001 general practices using a rapid antibody test. The study design, including sample size calculation, is described in the study protocol and summarized in our previous paper [20]. Briefly, participants eligible for this study included Belgian GPs and GP trainees practicing in primary care, as well as other PHCPs working in the same practice, such as nurses and physiotherapists. Eligible participants were required to have direct patient contact, the ability to adhere to the study’s protocol, and the capacity to provide informed consent.

This sub-study included only the PHCPs participating in the CHARMING study that had accepted to be recontacted for further research. In February–March 2023, the 3100 PHCPs participating in the CHARMING study, having accepted to be recontacted, were invited to answer questions about their views on a periodic COVID-19 booster vaccine for themselves and their patients. 

### 2.2. Ethics Approval and Consent to Participate

The CHARMING study was approved by Antwerp University Hospital Ethics Committee (Belgian reference number: 3002020000237) on the 16 November 2020. An amendment for the sub-study study was approved on the 31 January 2023 (reference number: 2023–5137). The study was conducted according to the approved protocol and the principles outlined in the Declaration of Helsinki. At the start of the wider national study (December 2020), each participant was informed of the goal of the study, the intended use of the collected data, and the pseudonymization of their data; all participants signed an online informed consent form. Participants did not receive any gift or financial reward for the time invested.

### 2.3. Data Collection

In the CHARMING study, participants completed a baseline questionnaire between 24 December 2020, and 15 January 2021, which collected various individual characteristics, including type of job, age, gender, practice size (with solo, duo, group, and big corresponding to one, two, maximum of seven, and more than seven employees in the practice, respectively) and location, which were retained for analysis in this paper.

In this sub-study, each participant completed an online survey using LimeSurvey version 3.22 (Hamburg, Germany), sharing their views on a periodic COVID-19 booster vaccine for themselves and their patients. The survey was conducted between 17/02/2023 and 07/03/2023. A single reminder was issued via the LimeSurvey platform ten days after the initial survey was launched. Each participant was provided with a link to their questionnaire where they had to enter their identification code. The questions were formulated based on the latest version of the Vaccine Confidence Index (VCI, February 2023). Several authors of this paper (with diverse professional profiles) were directly involved in the elaboration of the final questionnaire available in the Appendix A. In the first part of the survey, participants were asked if they had received a primary vaccination course or if they had received the third, fourth, and/or fifth booster dose since the last CHARMING testing period that took place between 13/12/2021 and 05/01/2022. Participants were also asked if they were vaccinated against influenza for the 2022/2023 season and if they were involved in prescribing, administering, or recommending vaccines. The section on views on a periodic COVID-19 booster vaccine included willingness and confidence toward periodic booster, likelihood to recommend a periodic booster, their views on their role in communication about a periodic booster, their views on who should receive or administer a periodic booster and favored and most trustful sources of information.

### 2.4. Data and Statistical Analysis

Descriptive statistics were conducted as follows: qualitative variables were reported as frequencies and percentages, while continuous variable (age) was presented using the median and interquartile range of 25th and 75th percentile (IQR 25 and 75).

In order to study the effects of various factors on willingness to get a periodic COVID-19 booster, a new binary variable was created for willingness to get a periodic booster vaccine with PHPCs who answered “Would you accept a periodic booster if it was an official recommendation and you had already had all previous vaccines?” with PHPCs who wish to receive a periodic booster (yes, definitely; unsure, but leaning toward yes) on one side and those who do not wish to receive a periodic booster (unsure, but leaning towards no; no, definitely not) on the other. For the variable “willingness to periodically recommend a COVID-19 booster vaccine to eligible patients”, we classified those who were likely to recommend it (highly likely and somewhat likely) on one side and those who were not likely (I do not know, somewhat likely, highly likely) on the other, based on the answer to “How likely are you to periodically recommend a COVID-19 booster vaccine to eligible patients?”. The odds ratios (OR) were estimated based on a logistic regression analysis. For this purpose, for all covariates, unadjusted odds ratios based on univariate analysis (uOR) and adjusted odds ratios based on multivariate analysis (aOR) and their 95% confidence interval (95% CI) were reported. The assumptions for logistic regression were examined with multicollinearity tested using the variance inflation factor (VIF) and excluded if VIF was under 4, and the influence of each observation on the overall regression model was tested using Cook’s Distance (Di) and considered as low influence if Di < 1. To check whether the data contained potential influential observations, the standardized residual error was inspected, and potential influential data points were then filtered. Independence of errors was respected as each participant was provided with a link to their questionnaire where they had to enter their identification code and the questionnaire could only be completed once by each participant [21]. In the first model (Model 1), sociodemographic information (age, gender, region, type of job, and practice size) were considered together. The second model (Model 2) shows multivariate analysis taking participants’ experiences into account: side effects related to the influenza vaccine, number of booster doses of COVID-19 vaccine received, confidence in vaccination, involvement in vaccination as a healthcare professional, as well as their vision of their role in encouraging vaccination and whether a booster should be administered to eligible patients. 

For vaccine confidence variable generation, PHCPs were considered as vaccine confident (Yes) if they agree or strongly agree with the three statements regarding the safety, importance, and effectiveness of boosters against the development of severe forms of COVID-19. 

Reasons for acceptance or refusal of a periodic booster for themselves or refusal to recommend a periodic booster to their patients were analyzed through multiple-choice questions where participants had to rank the three first items in order of preference. Reasons listed in results section are therefore not mutually exclusive.

The distribution of responses concerning privileged information and sources of trust is given as a percentage of PHCPs, using a 5-point Likert scale (for source of privileged information: strongly agree, tend to agree, I do not know, tend to disagree and strongly disagree/for source of trust: strongly agree, agree, neither agree nor disagree, disagree, strongly disagree).

Participants were required to answer each question before proceeding to the next question of the questionnaire. However, they could choose “prefer not to answer” or select “other” to provide an alternative answer. All statistical tests were two-sided, with a significant level set up at *p* ≤ 0.05. Statistical analyses and graphical representations were conducted using R (version 4.1.1, Vienna, Austria) and Microsoft Excel 2019 (Redmond, WA, USA).

### 2.5. Data and Qualitative Analysis

For the section regarding their views on who should administer a periodic booster, participants had the possibility to answer the following open question: “If you would suggest something else regarding who should administer the COVID-19 booster vaccine, please state it here:”. Suggestions made by participants were manually analyzed qualitatively using thematic content analysis. Qualitative analysis was performed by MD. The qualitative data were initially reviewed to gain a comprehensive understanding of their content. Subsequently, an inductive open-coding process was conducted using Microsoft Excel 2019 (Redmond, WA, USA) to create initial codes for data segments. Themes were determined based on this open coding, and they were further refined as we observed patterns, recurring concepts, and connections among the codes. General and more specific themes were attributed to each short answer. A list with themes is available in Appendix A .

## 3. Results

### 3.1. Participation Characteristics

Of the total cohort (N = 3100), 1900 (61%) PHCPs completed the survey about their views on a periodic COVID-19 booster vaccine for themselves and their patients. Among the 1900 respondents, 1814 answered the baseline questionnaire. Most respondents were over 40 years old (N = 1082; 59.6%), female (N = 1223, 67.4%), Flemish (N = 1299; 71.6%), worked as general practitioners (N = 1444; 79.6%), and in big practices (N = 733; 40.4%). PHCPs were mainly involved in prescribing, administrating, or recommending vaccines (N = 1551, 85.5%). Concerning the last influenza vaccination received, only 3.5% (N = 64) reported having mild to severe side effects. Concerning booster vaccination, the majority had received two booster doses since the last CHARMING testing period that took place between 13/12/2021 and 05/01/2022 (N = 1212, 66.8%) (Table 1). When comparing the sociodemographic information of PHCPs by type of job, we observe that there are consistently more women, workers from Flanders, and PHCPs working in big practices, regardless of the type of job (Appendix A).

### 3.2. Willingness to Get a Periodic COVID-19 Booster, Confidence in COVID-19 Booster Vaccine, and Association with Individual Characteristics

Among 1900 respondents, 1724 provided responses to the question, "Would you accept a periodic booster COVID-19 vaccine if it was an official recommendation and you had already received all previous vaccines?". Of these, a total of 1481 (86%) PHCPs would accept a periodic booster COVID-19 vaccine if it was an official recommendation (Figure 1). The top three reasons for acceptance of a periodic COVID-19 booster were “To protect themselves against getting seriously ill with COVID-19” (N = 905 in rank 1), “To protect their patients against getting seriously ill with COVID-19” (N = 876 in rank 2), and “They don’t want/wouldn’t want to get sick so they can provide consistency in the practice for patients and colleagues” (N = 790 in rank 3). For the 14% (N = 243) of participants who did not intend to get a periodic COVID-19 booster, the three main reasons for this refusal were that “They do not feel that COVID-19 presents a serious risk for them” (N = 105 in rank 1), “They have been seriously infected with COVID-19 and have natural immunity” (N = 88 in rank 2), and “They do not think that COVID-19 vaccines are effective at preventing getting infected with COVID-19” (N = 77 in rank 3).

Confidence in the safety, importance, and effectiveness of COVID-19 vaccines is the most widely informative determinant of vaccination intent [22]. Participants felt that COVID-19 boosters are safe (94%), important (90%), and effective against the development of severe forms of COVID-19 (95%), but fewer participants indicated that they are effective in preventing them from becoming infected (59%) and transmitting the virus (56%) (Appendix A).

To analyze which groups would be more or less likely to get a periodic COVID-19 booster vaccine for themselves, we compared the sociodemographic characteristics of participants who would accept a periodic COVID-19 booster with those who refused to get it (Table 2). PHCPs working in Brussels and Wallonia had a higher odd of unwillingness to get a periodic COVID-19 booster than their Flemish counterparts (uOR Wallonia 1.81, 95% CI:1.3, 2.50 and uOR Brussels 2.12, 95% CI: 1.32, 3.30). Non-GP or non-GP trainees, as well as PHCPs not involved in prescribing, administering, or recommending vaccines, had a higher odds of unwillingness to get a periodic COVID-19 booster than GPs and those involved in vaccination, respectively (uOR nurse 3.4, 95% CI: 1.71, 5.23; uOR physiotherapists 6.34, 95% CI: 3.16, 12.60; uOR other 2.7, 95% CI: 1.68, 4.24; and uOR PHCP not involved in vaccination 5.97, 95% CI: 3.85, 9.23). Male PHCPs were less likely to be unwilling to get a periodic booster than females (uOR male 0.6, 95% CI: 0.44, 0.83), and the more booster doses PHCPs had received since the last CHARMING survey, the less likely they were to hesitate to get periodic booster vaccines (uOR 1 dose 0.31, 95% CI: 0.15, 0.61; uOR 2 doses 0.03, 95% CI: 0.02, 0.06; and uOR 3 doses 0.00, 95% CI: 0.00, 0.02). PHCPs who did not trust vaccination had higher odds of unwillingness to get boosters (uOR for no confidence in vaccination 25.2, 95% CI: 16.70, 38.60) than PHCPs who trusted vaccination. Finally, those who stated that they do not completely agree that it is their role to encourage their patients to get a periodic booster or that the booster should be administered to all eligible patients were more likely to be unwilling to get a periodic booster compared to those who strongly agree or were highly likely, respectively (uOR tend to agree 5.01, 95% CI: 3.36, 7.71; uOR tend to disagree 12.50, 95% CI: 7.10, 22.10; uOR strongly disagree 131, 95% CI: 54.40, 370; uOR I do not know 22, 95% CI: 10.40, 46.80; uOR somewhat likely 6.95, 95% CI: 4.85, 10.10; uOR somewhat unlikely 73.00, 95% CI: 35.90, 160.00; uOR highly unlikely 528, 95% CI: 108.00, 9553.00; and uOR I do not know 19.1, 95% CI: 8.96, 40.50) (Table 2).

However, when considering the aOR for sociodemographic covariates (age, type of job, gender, practice size, and region) in Model 1, not being a GP or GP-trainee and practicing in Brussels or Wallonia remained associated with lower willingness to get a periodic COVID-19 booster vaccine compared to being a GP and their Flemish counterparts, respectively (aOR nurse 3.86, 95% CI: 1.98, 7.28; aOR physiotherapists 5.48, 95% CI: 2.28, 12.16; aOR other 2.78, 95% CI: 1.55, 4.80; aOR Wallonia 2.02, 95% CI:1.35, 2.99; and aOR Brussels 2.7, 95% CI: 1.58, 4.50). The willingness to get a periodic COVID-19 booster did not vary with age when considered as a continuous variable. However, when age was categorized (≤30, 31–40, 41–50, 51–60, and >60), participants in the 31–40 and 41–50 age groups were more likely to be unwilling to get a periodic booster compared to participants aged 30 or under (aOR 31–40 2.52, 95% CI: 1.34, 5.09 and aOR 41–50 2.46, 95% CI: 1.27, 5.08) (Appendix A). When we consider together the participants’ experiences (side effects related to the influenza vaccine, the number of booster doses of COVID-19 vaccine received, the confidence in vaccination, the involvement in vaccination as a healthcare professional, as well as their vision of their role in encouraging vaccination and if a booster should be administered to eligible patients) in Model 2, having received one booster is no longer associated with a smaller odds of unwillingness to get a periodic booster, while having received two or three boosters remained associated with a smaller odds of unwillingness to get a periodic booster vaccine compared with those not having received any dose since the last CHARMING survey (aOR one dose 0.86, 95% CI: 0.20, 4.18; aOR two doses 0.18, 95% CI: 0.04, 0.87; and aOR three doses 0.04, 95% CI: 0.04, 0.00, 0.48). PHCPs who did not have confidence in vaccination still had higher odds of unwillingness to get a periodic booster compared to those who had confidence (aOR no confidence in vaccination 5.62, 95% CI: 2.68, 11.7) while not being implicated in vaccination as a healthcare provider is no longer associated (aOR PHCP not involved in vaccination 1.14, 95% CI: 0.32, 3.44). Finally, PHCPs who answered not being highly likely to periodically recommend a COVID-19 booster vaccine to eligible patients had higher odds of unwillingness to get a periodic booster compared to those being highly likely (aOR somewhat likely 3.12, 95% CI: 1.71, 5.76; aOR somewhat unlikely 19.90, 95% CI: 5.42, 76.70; aOR highly unlikely 25.3, 95% CI: 1.15, 1380.00; and aOR I do not know 19.5, 95% CI: 4.11, 77.90). Those who say it is definitely not their role to encourage eligible patients to get a booster are also associated with higher odds of unwillingness to get a booster compared to those who strongly agree when asked if it is their role to encourage eligible patients to receive a booster (aOR strongly disagree 29.3, 95% CI: 4.59, 225) (Table 2).

### 3.3. Target Population for Periodic Booster Vaccines and Likelihood of Recommending a COVID-19 Booster Vaccine and Principal Reasons for Not Recommending It

Participants believed that the groups suitable for administering booster vaccines are similar to those for administering influenza vaccines, encompassing high-risk patients, high-risk PHCPs, and PHCPs in general while excluding the general population (Appendix A).

Although 243 participants would not take the booster periodically, only 74 would not recommend it (2% do not know and 4% highly or somewhat unlikely) (Appendix A). The main reasons for non-recommending a periodic booster were that “The approval/development of the vaccine and boosters may be rushed, and the vaccine may not have been sufficiently tested”, “They do not know enough to recommend it”, and “the vaccine could have serious side effects”.

When analyzing the impact of various sociodemographic factors (like age, job type, gender, practice size, and region) on the unwillingness to recommend periodic booster vaccines through multivariate analysis, we found that physiotherapists, nurses, and professionals other than GP trainees were more likely to not recommend it compared to GPs. Additionally, practicing in Brussels or Wallonia was also linked to higher odds of unwillingness to recommend booster vaccines. Interestingly, factors like age, gender, and practice size did not show any significant influence on the odds of recommending these boosters.

When we took into account participants’ experiences, including side effects of the influenza vaccine, the number of COVID-19 booster doses they received, their confidence in vaccination, their involvement in vaccination as healthcare professionals, as well as their personal willingness to get a booster and their attitude to recommending boosters to eligible patients, we found that a lack of confidence in vaccination, personal unwillingness to get periodic booster vaccines, and primary healthcare professionals who firmly believe it is not their role to encourage eligible patients for boosters were also associated with a higher odds of unwillingness in recommending them (Appendix A).

### 3.4. Perspectives on HCP in Charge of Administering a COVID-19 Booster Vaccine

When it comes to determining which professions should be responsible for administering boosters, the surveyed population had mixed views about the extent to which they would support COVID-19 booster vaccine administration by pharmacists. Out of the 1723 participants that answered the question, 59% (N = 1012) stated that they “strongly agree” and “tend to agree”, while 39% (N = 667) “tend to disagree” or “strongly disagree”, and 2% (N = 44) did not know. When asked whether they believed that vaccination by pharmacists would reduce their workload, 42% (N = 723) did not think it would reduce their workload, 45% (N = 777) considered that it would reduce their workload, and 13% (N = 223) did not know.

Overall, 308 PHCPs provided responses to the question regarding “Who should administer the COVID-19 booster vaccine”. Out of the 308 responses received from PHCPs, 290 were relevant to the question and were included in our analysis. Following a manual qualitative analysis, some PHCPs suggested a specific HCP to administer the vaccine, while others recommended the establishment of a structure or organization, with or without a designated individual for vaccine administration. In cases where PHCPs proposed GPs, nurses, or pharmacists to administer booster vaccines, some specified certain conditions related to the work environment (e.g., prevention center, general practice, occupational medicine) and mentioned collaboration with other specific HCPs (such as nurses or pharmacists under GP supervision). They also provided additional indications about conditions, such as preferring administration during the influenza vaccine delivery or mentioning the advantage of single-dose products with less stringent storage requirements. Some PHCPs simply emphasized the importance of collaboration without specifying who should perform the vaccination. We also observed that some indicated they did not prefer the GPs to administer the boosters while suggesting collaborative or structural alternatives or explaining why it is not feasible for GPs.

### 3.5. Sources of Privileged Information and Sources of Trust

The majority stated that they would not find reliable information about COVID-19 from a complementary or alternative medical practitioner (77%) or from the internet or social media (65%). Opinions were more mixed regarding information from media (53%). The majority stated that the government (85%), health professionals (86%), health authorities (87%), and international organizations (87%) provide reliable information about COVID-19. Compared to before the COVID-19 pandemic, 54% of PHCPs gained more confidence in the importance of vaccination, 49% in the effectiveness of vaccines, and 44% in the safety of vaccines. Overall, 61% felt more informed about vaccination in general compared to before the COVID-19 pandemic. Concerning the government, 76% were satisfied with the way the government has handled COVID-19 vaccination, and 72% considered that the government’s actions were in their personal best interests. Finally, 89% strongly agreed or tended to agree that the government is committed to protecting the public from COVID-19.

Concerning trust in authorities’ recommendations, 46%, 41%, and 40% neither agreed nor disagreed with the following statements: “I feel/felt that the official authorities understand/understood my needs”, “I feel/I felt the official authorities are/were cold and distant”, and “I feel/felt that the official authorities care about me”, respectively. The majority strongly disagreed or disagreed when asked if: they feel/felt excluded by official authorities (54%), they would be punished if they did not get vaccinated (55%), they feel/felt coerced to get vaccinated (66%), they have/had serious doubts about whether they could get vaccinated if they wanted (83%), and it would be difficult for them to get vaccinated if they wanted to (86%). On the other hand, they mainly strongly agreed or agreed when asking if they feel/felt a sense of choice and freedom in whether to get a vaccine (56%), they feel/felt that their decision to vaccinate reflects what they really want (69%), and they feel/felt confident and capable that they could get vaccinated if they wanted to (85% and 87%, respectively).

## 4. Discussion

This study reports a high level of willingness (86%) among PHCPs to accept periodic COVID-19 booster vaccines if officially recommended. This is an optimistic signal, illustrating a resolute commitment to ensuring the safety and health of both them and their patients. While vaccine acceptance may fluctuate depending on the circumstances and time frame, our findings obtained in February–March 2023 are similar to those of a study conducted in the USA in February–March 2021 among 1374 HCPs. That study revealed that 83.6% of HCPs were open to receiving an annual booster [17]. Additionally, other studies in Poland, Singapore, and the UK have indicated that approximately 75% of HCPs are either willing or not hesitant to get a booster [23,24,25].

PHCPs generally expressed high levels of confidence in the safety, importance, and effectiveness of COVID-19 vaccines. This is crucial for promoting vaccination uptake, as HCPs often play a significant role in guiding patient decisions [9,10]. This study identifies several factors associated with unwillingness among PHCPs to receive a booster, including not being a GP or a GP trainee and working in certain regions. Several studies have previously noted that nurses express a less favorable opinion of the influenza vaccine when compared to physicians [26]. Regarding COVID-19 vaccination, in March 2021, the willingness to get vaccinated differed among various professions, with doctors exhibiting the highest vaccination rates, at 75%, while a cohort of nurses and nursing aides had the lowest acceptance rate, at 56.7% and 45.6%, respectively [27]. Moreover, the disparities we identified in our study among various Belgian regions were consistent with the vaccination coverage observed among HCPs in October 2021 [28]. Full vaccination coverage ranged from 72.9% for HCPs living in Brussels and 83.3% in Wallonia to 94.7% for HCPs residing in Flanders [29]. Additionally, in our prior study of vaccine hesitancy among nursing home staff members (NHS) in Belgium, it was observed that NHSs in Wallonia exhibited a higher propensity for vaccine hesitancy in contrast to their Flemish counterparts, with a 2.22-fold increased likelihood of hesitancy toward vaccination [30]. Regional disparities in periodic COVID-19 vaccine boosters’ acceptance and HCP roles are evident in the findings. PHCPs in Brussels and Wallonia were more likely to be unwilling to receive boosters. In addition, no gender differences or differences between the type of practice were demonstrated in the present study. This suggests that tailoring vaccination strategies to specific regions and healthcare roles may be necessary. We also confirmed that measuring confidence in vaccination, as carried out in this study, is a determinant of intention to receive a vaccine [22].

Despite the general optimism, it is important to address the 14% of PHCPs who did not intend to get periodic booster vaccines for themselves. As introduced earlier, even though it may not be prevalent, vaccine hesitancy among HCPs exists [14]. Some studies have indicated that vaccine hesitancy is strongly associated with negative emotions such as anger and cynicism. Although they represent a minority, these groups could potentially undermine vaccine booster campaigns, even after the pandemic [31]. The ongoing prevalence of anti-establishment beliefs, encompassing conspiracy theories and a widespread lack of confidence in institutions, significantly influences vaccine acceptance. This underscores a challenge: effectively addressing vaccine hesitancy demands not only the spread of correct information but also the restoration of trust in medical establishments and tackling the deeper socio-political reasons behind this skepticism [32].

It is noteworthy that a smaller number of PHCPs were unwilling to recommend periodic boosters to their patients compared to those unwilling to receive them personally. This underscores the complex decision-making process among HCPs, who consider different factors when advising patients [33]. Moreover, although this is in the direction of promoting vaccination for their patients, even if not for them, while it may seem strange that HCPs do not follow their own recommendations, this is nothing new [34]. In our study, among reasons for non-recommending it, PHCPs expressed that the approval/development of the vaccine and boosters may be rushed, and the vaccine may not have been sufficiently tested. We also saw in this study that a lack of confidence in COVID-19 vaccination was a predictor of unwillingness to recommend periodic booster vaccination. There is evidence that HCPs often express a desire for more substantial and robust evidence, both in terms of quality and quantity, when making decisions about vaccinations and whether to recommend them [35,36,37]. Evidence-based communication is vital for building HCPs’ trust and their ability to recommend vaccination, underscoring the need for ongoing training programs focused on vaccine research, safety data, and effective communication strategies to maintain consistent recommendations, as already suggested [38]. Until 2023, pharmacists in Belgium were not allowed to administer vaccines, unlike some other countries, where they were authorized to vaccinate even before the pandemic, enabling them to start vaccinating against COVID-19 very early [19,39,40]. A study conducted during the initial phase of the COVID-19 pandemic highlighted that pharmacists exhibited both the capacity and the willingness to adjust existing clinical services and introduce innovative approaches to address the recurring waves of the pandemic [40]. Furthermore, recent studies provided evidence in favor of incorporating pharmacies as an added resource for administering COVID-19 vaccines [41,42]. In our study, the mixed opinions of PHCPs on whether pharmacists should administer COVID-19 boosters highlight the need for clear guidelines and communication about the roles of different healthcare professionals in the vaccination process, including in recording the vaccination administered by pharmacists to maintain informational continuity of care. This could help to standardize vaccine administration and reduce confusion. As shown in a 2014 study on the opinions of pharmacists and general practitioners in Ireland regarding the expansion of the community pharmacist’s role, it becomes evident that effective communication between different stakeholders remains the key to clarifying each party’s role [43]. In the current study, various proposals made by the 290 PHCPs, including the administration of COVID-19 booster vaccines by specific HCPs such as GPs, nurses, or pharmacists or the need for organizational structures and collaborative approaches, highlight the complexity of the subject, the need for communication, and the joint development of such recommendations.

This study indicates that PHCPs generally trust traditional healthcare sources and government agencies for COVID-19 information. A significant proportion does not trust complementary or alternative practitioners or information from the internet and social media. This is in line with a study conducted among 2683 American HCPs. HCPs most frequently trusted government agencies and not social media for COVID-19 vaccine information [44].

The strengths of this study include that this national study is the first in Belgium to evaluate the views of a large cohort of PHCPs on periodic COVID-19 booster vaccination for themselves and their patients, the reasons for acceptance or refusal, confidence in vaccine safety and efficacy, as well as views on booster recommendations. In addition, a mixed method using quantitative and qualitative analysis of open-ended questions enabled this study to go further in understanding the views of PHCPs on the question of who should administer a periodic booster. Finally, this sub-study included only the PHCPs participating in the CHARMING study who accepted to be recontacted for further research. Participants were therefore familiar with the questionnaires and study team, which may have limited the social desirability bias due to the nature of this study, funded by the Scientific Institute of Public Health (SCIENSANO).

This study also has certain limitations, including potential selection bias, because it relies on self-reported data from a convenience sample of PHCPs who accepted to be recontacted for further research. The findings may, therefore, not fully represent the PHCP population’s attitudes toward vaccination. In addition, eligible participants for this study included Belgian GPs and GP trainees practicing in primary care, as well as other PHCPs working in the same practices, such as nurses and physiotherapists. However, while PHCPs with another type of job than GP are well represented in our sample of PHCPs working with a GP in the same practice, nurses and physiotherapists were under-represented relative to their respective populations in Belgium, so our findings cannot be generalized to these groups. This study also highlights differences in attitudes and willingness to get booster vaccines based on the region of practice (Flanders, Wallonia, and Brussels). However, it does not explore the underlying reasons for these regional disparities, which could be important for public health interventions. Finally, the present study incorporates some qualitative data analysis in the section on perspectives about the authority in charge of administering booster vaccines. However, it might be beneficial to include more qualitative data analysis to gain a deeper understanding of participants’ perspectives and reasons behind their attitudes. This type of qualitative study was conducted among healthcare workers in Canada to identify factors associated with the acceptance of influenza and COVID-19 vaccines. Overall, study participants indicated that factors such as evidence-based decision-making, a sense of responsibility, accessibility, normative influences, and policy considerations influenced their decision to get vaccinated against COVID-19 and influenza. This article also offers recommendations for shaping future interventions and messages based on these factors. Overcoming barriers, future messages, and interventions may include organizing vaccination campaigns to better raise awareness about the health consequences of these pathogens, providing more scientific information to healthcare professionals to enhance their capacity in recommending specific vaccines, maximizing vaccine access in workplaces, and complementing vaccination requirements with educational efforts to increase support for these policies [45]. In a study conducted in France, a neighboring country, similar findings were suggested. Following qualitative research in a group of French HCPs, it was recommended that disseminating clear and comforting scientific information by HCPs and vaccine experts, tailored to local situations, could enhance HCPs’ intentions to receive COVID-19 vaccines [46]. An ongoing randomized controlled trial is focusing on vaccine-averse populations of healthcare workers, to better understand how to improve their vaccine confidence and information dissemination. In this randomized study, researchers will evaluate the effects of three interventions on confidence in COVID-19 vaccination [47].

Our findings have policy implications for public health authorities. Tailored communication campaigns and educational interventions may be needed to address vaccine hesitancy and encourage booster uptake among specific groups, such as non-GP HCPs or those in certain regions.

## 5. Conclusions

In conclusion, this study provides valuable insights into the views of PHCPs regarding periodic COVID-19 booster vaccines. The majority of PHCPs support these booster vaccines, particularly GPs and GP trainees, who are more likely to recommend and receive them compared to other PHCPs. However, notable hesitancy exists among non-GPs and those working in Wallonia and Brussels. Furthermore, mixed opinions among PHCPs regarding the administration of COVID-19 booster vaccines indicate the importance of establishing clear guidelines and effective communication regarding the roles of various healthcare professionals in the vaccination process.

Overall, these findings highlight the need for targeted communication and educational initiatives to address vaccine hesitancy and promote booster uptake. Strategies should consider regional and professional differences. Finally, understanding the reasons behind the varying acceptance levels can inform public health strategies aimed at increasing vaccination uptake, addressing vaccine hesitancy, and ensuring that PHCPs play a key role in promoting vaccination among their patients, even after the pandemic.

## Figures and Tables

**Figure 1 vaccines-12-00740-f001:**
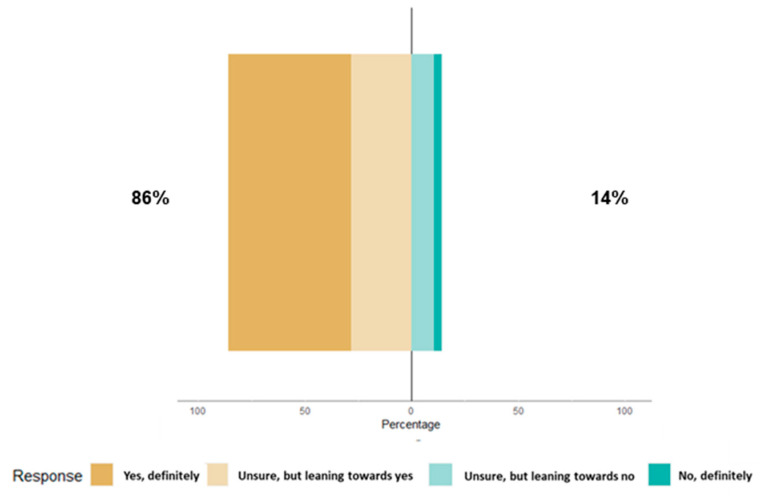
Willingness to accept periodic COVID-19 booster vaccines if it was an official recommendation and primary healthcare providers (PHCPs) who had already received all previous vaccines.

**Table 1 vaccines-12-00740-t001:** Characteristics of 1814 Belgian primary healthcare providers (PHCPs) responding to the survey about their views on periodic COVID-19 booster vaccine for themselves and their patients (between 17/02/2023 and 07/03/2023).

	N	%		N	%		N	%
**Gender**			**Side effect related to influenza vaccination**			**Number of boosters received**		
Male	591	32.6	No effect	1014	55.9	0	108	5.9
Female	1223	67.4	Negligible	312	17.2	1	286	15.8
**Region**			Mild	55	3.0	2	1212	66.8
Flanders	1299	71.6	Moderate	8	0.4	3	114	6.3
Wallonia	373	20.6	Severe	1	0.05	Other	94	5.2
Brussels	142	7.8	I do not remember	2	0.15	**Confidence in vaccination**		
**Type of job**			Unknown	422	23.3	No	377	20.8
General practitioner	1444	79.6	**Implication in vaccination as a healthcare worker**			Yes	1437	79.2
Nurse	76	4.2	Yes	1551	85.5	**Healthcare professional’s views of their role in encouraging eligible patients to receive a booster**		
Physiotherapist	36	2.0	No	94	5.2	Strongly agree	768	42.7
Training GP	56	3.1	Unknown	169	9.3	Tend to agree	709	39.0
Other	115	6.3	**Practice size**			Tend to disagree	92	5.1
Unknown	87	4.8	Solo	372	20.5	Strongly disagree	38	2.0
**Healthcare professional’s views regarding if booster should be administered to eligible patients**			Duo	288	15.9	I do not know	36	1.9
Highly likely	1054	58.1	Group	397	21.9	Unknown	171	9.3
Somewhat likely	486	26.8	Big	733	40.4			
Somewhat unlikely	46	2.5	Unknown	24	1.3			
Highly unlikely	24	1.3						
I do not know	33	1.8						
Unknown	171	9.5						

**Table 2 vaccines-12-00740-t002:** Odds of unwillingness to get a periodic COVID-19 booster vaccine as a function of individual characteristics (N = 1644). Bold numbers highlight the results statistically significant.

	Willingness to Get a Periodic Booster	Unwillingness to Get a Periodic Booster	Unadjusted OR [95% CI]	Adjusted OR [95% CI]
	N = 1414	N = 230		
				**Model 1**
**Age (years)**			0.99 [0.98, 1.00]	0.99 [0.97, 1.00]
Median (IQR)	*46*	*43*		
**Gender**				
Female (ref)	915	173	1	1
Male	499	57	**0.6** **[0.44, 0.83]**	0.8 [0.55, 1.17]
**Region**				
Brussels	100	28	**2.12** **[1.32, 3.30]**	**2.7** **[1.58, 4.5]**
Flanders (ref)	1051	139	1	1
Wallonia	263	63	**1.81** **[1.30, 2.50]**	**2.02** **[1.35, 2.99]**
**Type of job**				
General practitioner (ref)	1227	163	**1**	1
GP trainee	43	4	0.7 [0.21, 1.76]	0.7 [0.2, 1.85]
Nurse	47	19	**3.4** **[1.71, 5.23]**	**3.86** **[1.98, 7.28]**
Physiotherapist	19	16	**6.34** **[3.16, 12.60]**	**5.48** **[2.28, 12.6]**
Other	78	28	**2.7** **[1.68, 4.24]**	**2.78** **[1.55, 4.8]**
**Practice size**				
Solo (ref)	298	41	1	1
Duo	220	38	1.26 [0.78, 2.20]	1.21 [0.69, 2.14]
Group	314	53	1.23 [0.79, 1.91]	1.67 [0.99, 0.86]
Big	563	93	1.2 [0.82, 1.79]	0.88 [0.54, 1.45]
NA	19	5	NA	NA
				**Model 2**
**Side effects related to influenza vaccination**				
No effect (ref)	889	77	1	1
Negligible	284	20	0.82 [0.48, 1.34]	0.78 [0.40, 1.47]
Mild	47	7	1.74 [0.7, 3.75]	1.11 [0.3, 3.37]
Moderate	5	2	4.67 [0.66, 2.21 × 10^1^]	0.07 [0, 2.37]
Severe	1	0	0.00 [NA, 7.45 × 10^72^]	0.00 [NA, 4.06 × 10^123^]
I do not remember	2	0	NA	NA
NA	186	124	NA	NA
**Number of boosters received since last CHARMING testing**				
Zero	37	60	1	1
One	156	97	**0.31** **[0.15, 0.61]**	0.86 [0.2, 4.18]
Two	1105	71	**0.03** **[0.02, 0.06]**	**0.18** **[0.04, 0.87]**
Three	112	1	**0.00** **[0.00, 0.02]**	**0.04** **[0, 0.48]**
Other	4	1	NA	NA
**Confidence in vaccination**				
Yes (ref)	1336	101	1	1
No	78	129	**25.2** **[16.70, 38.60]**	**5.62** **[2.68, 11.7]**
**Implication in vaccination as a healthcare worker**				
Yes (ref)	1363	188	1	1
No	51	42	**5.97** **[3.85, 9.23]**	1.14 [0.32, 3.44]
**Healthcare professional’s views of their role in encouraging eligible patients to receive a booster**				
Strongly agree (ref)	738	30	1	1
Tend to agree	589	120	**5.01** **[3.36, 7.71]**	**2.19** **[1.14, 4.36]**
Tend to disagree	61	31	**12.5** **[7.10, 22.10]**	1.05 [0.93, 3.37]
Strongly disagree	6	32	**131** **[54.40, 370]**	**29.3** **[4.59, 225]**
I do not know	19	17	**22.0** **[10.40, 46.80]**	0.49 [0.04, 3.6]
NA	1	0	NA	NA
**Healthcare professional’s views regarding if booster should be administered to eligible patients**				
Highly likely (ref)	1010	44	1	1
Somewhat likely	373	113	**6.95** **[4.85, 10.1]**	**3.12** **[1.71, 5.76]**
Somewhat unlikely	11	35	**73.00** **[35.90, 160.00]**	**19.9** **[5.42, 76.7]**
Highly unlikely	1	23	**528** **[108.00, 9553.00]**	**25.3** **[1.15, 1380]**
I do not know	18	15	**19.1** **[8.96, 40.50]**	**19.5** **[4.11, 77.9]**
NA	1	0	NA	NA

Willingness vs. unwillingness to get a periodic COVID-19 booster vaccine was determined as follows: PHCP who reported “yes, definitely” or “unsure, but leaning towards yes” to the question “Would you accept a periodic booster COVID-19 vaccine if it was an official recommendation and you had already had all previous vaccines?” were determined as willing to get a periodic booster. PHCP who answered “unsure, but leaning towards no” and “no, definitely” were considered as unwilling to get it. Profiles are distributed by individual characteristics in the first model (Model 1) as age (median); gender; region; and type of job, with jobs divided into general practitioners (GPs), GP trainees, nurses, physiotherapists, and other jobs, and practice size (with solo being one employee, duo being two employees, group practice being maximum of seven employees, and big being more than seven employees in the practice). In the second model (Model 2), participants’ experiences are taken into account: self-reported side effects related to last influenza vaccination; number of boosters received since the last CHARMING testing that took place on 13/12/2021 and 05/01/2022; confidence in vaccination based on positive responses for the three statements concerning the safety, importance, and efficacy of booster doses against the development of severe forms of COVID-19; implication in vaccination as a healthcare worker; PHCP’s views of their role in encouraging eligible patients to receive a booster when asking the question, “Do you believe that it is your role to encourage your patients to get vaccinated even if they are hesitant ?”; and their views on the question, “How likely are you to periodically recommend a COVID-19 booster vaccine to eligible patients?”. Data are shown as unadjusted odds ratios (ORs) with 95% confidence intervals (95% CIs) and adjusted OR for all covariates (multivariate analysis) with 95% CI. The ORs are estimated based on a logistic regression analysis.

## Data Availability

Data is contained within the article or Appendix A. The original contributions presented in the study are included in the article/Appendix A, further inquiries can be directed to the corresponding author.

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
