# Peer review of "Primary Healthcare Providers’ Views on Periodic COVID-19 Booster Vaccination for Themselves and Their Patients: A 2023 Nationwide Survey in Belgium"

_vaccines, 2024, doi:10.3390/vaccines12070740_

Round 1
Reviewer 1 Report
Comments and Suggestions for Authors
I was invited to revise the paper entitled "Primary healthcare providers' views on periodic COVID-19 booster vaccination for themselves and their patients: a 2023 nationwide survey in Belgium". It aimed to evaluate vaccine hesitancy and associated factors among primary HC physician in Belgium.
It was a sub-study of a large cohort performed in Belgium during pandemic.
The topic is relevant for public health and poor study were conducted in Belgium on this topic among PHCP.
Observations:
- I reccomend to categorize age by age classes in order to evaluate the change in willingness to achieve booster dose by age; It is well knoewn that age is a significant factor associated to VH;
- Last two variables in table 2 were presented as Likert Scale. I suggest to present them also as median and IQR;
- I suggest to add comparisons of baseline characteristics by type of job reported in table 2;
- Strenght and limitation section was lacking. I suggest to add it;
Author Response
I was invited to revise the paper entitled "Primary healthcare providers' views on periodic COVID-19 booster vaccination for themselves and their patients: a 2023 nationwide survey in Belgium". It aimed to evaluate vaccine hesitancy and associated factors among primary HC physician in Belgium.
It was a sub-study of a large cohort performed in Belgium during pandemic.
The topic is relevant for public health and poor study were conducted in Belgium on this topic among PHCP.
We thank Reviewer #1 for their review of our study and for providing valuable comments and suggestions. As detailed below, we have addressed all their comments and concerns, including the incorporation of suggested strength and limitation in the discussion section.
Observations:
- I reccomend to categorize age by age classes in order to evaluate the change in willingness to achieve booster dose by age; It is well knoewn that age is a significant factor associated to VH;
We thank Reviewer #1. Continuous variable categorization can indeed be done in logistic regression analysis. However, keeping continuous variables without categorizing them, prevent from information loss and keep granularity, allowing the model to detect subtler trends and relationships. In addition, continuous variables generally provide more statistical power than categorical ones. The choice of “cutpoints” for categorizing continuous variables may also introduce bias [1–3]. Based on these statements, variable “age” was kept as a continuous variable in our analysis.
- Ranganathan, P.; Pramesh, C.; Aggarwal, R. Common Pitfalls in Statistical Analysis: Logistic Regression. Perspect Clin Res 2017, 8, 148, doi:10.4103/picr.PICR_87_17.
- Naggara, O.; Raymond, J.; Guilbert, F.; Roy, D.; Weill, A.; Altman, D.G. Analysis by Categorizing or Dichotomizing Continuous Variables Is Inadvisable: An Example from the Natural History of Unruptured Aneurysms. AJNR Am J Neuroradiol 2011, 32, 437–440, doi:10.3174/ajnr.A2425.
- Bennette, C.; Vickers, A. Against Quantiles: Categorization of Continuous Variables in Epidemiologic Research, and Its Discontents. BMC Med Res Methodol 2012, 12, 21, doi:10.1186/1471-2288-12-21.
- Last two variables in table 2 were presented as Likert Scale. I suggest to present them also as median and IQR;
We thank Reviewer #1. Last two variables in Table 2 are “Healthcare professional's views of their role in encouraging eligible patients to receive a booster” and “Healthcare professional's views regarding if booster should be administered to eligible patients” both are categorical variables with categories being “Strongly agree”, “Tend to agree”, “Tend to disagree”, “Strongly disagree” and “I don’t know” and “Highly likely”, “Somewhat likely”, “Somewhat unlikely”, “Highly unlikely” and “I don’t know”, respectively. Categorical variables, by nature, represent distinct groups or categories and do not have a meaningful order or distribution that can be summarized by median and interquartile range (IQR). Especially the category "I don't know" prevents the use of summary statistics that could be applied to an otherwise ordinal variable. For categorical variables, appropriate descriptive statistics typically include:
- Frequencies (Counts): The number of observations in each category.
- Percentages: The proportion of observations in each category relative to the total number of observations.
In our analysis, these variables were described using counts and percentages.
- I suggest to add comparisons of baseline characteristics by type of job reported in table 2;
We thank Reviewer #1. To see how we have addressed this comment, please see the attached Table (TableS1) where we compared baseline characteristics by type of job. We observe that there are still more women, workers from Flanders and PHCPs working in big practices, regardless of the type of job. Within each group of PHCPs by job type, we maintain the distribution of socio-demographic characteristics (age, gender, etc.) as observed in the overall study population. In the logistic regression in Table 2 all variables shown in the table below are included and controlled for.
To clarify our study population, this Table comparing baseline characteristics by type of job was included in the supplementary materials as Table S1 and the Results section of the revised version of the manuscript now included the following text on population clarification:
“When comparing the sociodemographic information of PHCPs by type of job, we observe that there are consistently more women, workers from Flanders, and PHCPs working in big practices, regardless of the type of job (Table S1).” [line 209-line 212]
- Strenght and limitation section was lacking. I suggest to add it;
We thank Reviewer #1 and we have clarified the "strengths and limitations" section by adding a section on the strengths of our study and clarifying the limitations concerning the lack of representativeness of our population. In our study, non-GPs PHCPs, e.g. nurses and physiotherapists, are under-represented relatives to the respective populations of nurses and physiotherapists in Belgium and our findings cannot be generalized to the latter populations. However, we consider PHCPs with another type of job than GP to be well represented in our sample of PHCPs working with a GP in the same practice and our findings to be generalizable to PHCPs working in a general practice. The Discussion section of the revised version of the manuscript now included the following text on the strengths and the limitations of our study.
“The strengths of this study include that this national study is the first in Belgium to evaluate views of a large cohort of PHCPs on periodic COVID-19 booster vaccination for themselves and their patients, the reasons for acceptance or refusal, confidence in vaccine safety and efficacy as well as views on booster recommendations. In addition, a mixed method using quantitative and qualitative analysis of open-ended questions enabled this study to go further in understanding the views of PHCPs on the question of who should administer a periodic booster. Finally, this sub-study included only the PHCPs participating in the CHARMING study that accepted to be recontacted for further research. Participants were therefore familiar with the questionnaires and study team which may have limited the social desirability bias due to the nature of this study, funded by the scientific institute of public health (SCIENSANO).
This study has also certain limitations, including potential selection bias because it relies on self-reported data from a convenience sample of PHCPs that accepted to be recontacted for further research. The findings may, therefore, not fully represent the PHCP population's attitudes toward vaccination. In addition, eligible participants for this study included Belgian GPs and GP trainees practicing in primary care, as well as other PHCPs working in the same practices, such as nurses and physiotherapists. However, while PHCPs with another type of job than GP are well represented in our sample of PHCPs working with a GP in the same practice, nurses and physiotherapists were underrepresented relative to their respective populations in Belgium, so our findings cannot be generalized to these groups. The study also highlights differences in attitudes and willingness to get booster vaccines based on the region of practice (Flanders, Wallonia, Brussels). However, it does not explore the underlying reasons for these regional disparities, which could be important for public health interventions. Finally, the present study incorporates some qualitative data analysis in the section on perspectives about the authority in charge of administering booster vaccines. However, it might be beneficial to include more qualitative data analysis to gain a deeper understanding of participants' perspectives and reasons behind their attitudes. [line 482-line 509]

Reviewer 2 Report
Comments and Suggestions for Authors
The article is well structured, but its level of originality and novelty is not very high.
The conclusions of the article should better represent the findings of the study. As they are presented, they are highly unspecific.
Author Response
The article is well structured, but its level of originality and novelty is not very high.
We thank Reviewer #2 for their review of our study and for providing valuable comments and suggestions. As detailed below, we have addressed the comment regarding the conclusion to better represent findings of the study.
The conclusions of the article should better represent the findings of the study. As they are presented, they are highly unspecific.
We thank Reviewer #1 and we have adapted the conclusion to better represent the findings of the study. The Conclusion section of the revised version of the manuscript now included the following text :
“In conclusion, the study provides valuable insights into the views of PHCPs regarding periodic COVID-19 booster vaccines. The majority of PHCPs support these booster vaccines, particularly GPs and GP trainees, who are more likely to recommend and receive them compared to other PHCPs. However, notable hesitancy exists among non-GPs and those working in Wallonia and Brussels. Furthermore, mixed opinions among PHCPs regarding the administration of COVID-19 booster vaccines indicate the importance of establishing clear guidelines and effective communication regarding the roles of various healthcare professionals in the vaccination process.
Overall, these findings highlight the need for targeted communication and educational initiatives to address vaccine hesitancy and promote booster uptake. Strategies should consider regional and professional differences. Finally, understanding the reasons behind the varying acceptance levels can inform public health strategies aimed at increasing vaccination uptake, addressing vaccine hesitancy, and ensuring that PHCPs play a key role in promoting vaccination among their patients, even after the pandemic.” [line 534-line 547]
Reviewer 3 Report
Comments and Suggestions for Authors
I read the study by Marina Digregorio et.al entitled "Primary healthcare providers' views on periodic COVID-19 booster vaccination for themselves and their patients: a 2023 nationwide survey in Belgium". This is an interesting study, which highlights "the need to address vaccine confidence, regional disparities, and PHCP roles in booster implementation".
Here are my observations:
2. Materials and Methods
I am sorry but the proposal: "The study design including sample size calculation is described in the study protocol and summarized in our previous paper [19]" cannot be accepted. I would ask the authors to provide this study with details about its design, where at least they should explain:
-If the participants were a convenience sample.
- The adequacy and power of the sample
- The representativeness of the sample.
Particularly in terms of representativeness, the sample seems to have a significant problem, for example, nurses are probably under-represented. Also, could the researchers support the representativeness of the sample in terms of gender, age, and type of work compared to the general PHCP population?
Data and statistical analysis
In addition to the VIF, were the other assumptions for logistic regression such as independence of errors examined? Do you include the VIF value somewhere in the text?
4. Discussion
In the discussion the researchers present their findings from an optimistic perspective, underestimating a strong anti-vaccination minority that unfortunately also exists in Belgium. However, one year after the end of the pandemic crisis, the debate on the acceptance of the COVID-19 vaccine would be outmoded if we were not concerned with this minority. Post-pandemic study identifies a link between vaccination hesitancy, anger, cynicism, and distrust in the medical system. The problem is whether behind the non-acceptance of the vaccine are hidden wrong and destructive anti-systemic views of distrust in the medical system, which will remain active even after the pandemic. Please, the authors should address this concern in the discussion.
Add limitations of the study.
Author Response
I read the study by Marina Digregorio et.al entitled "Primary healthcare providers' views on periodic COVID-19 booster vaccination for themselves and their patients: a 2023 nationwide survey in Belgium". This is an interesting study, which highlights "the need to address vaccine confidence, regional disparities, and PHCP roles in booster implementation".
We thank Reviewer #3 for their review of our study and for providing valuable comments and suggestions. As detailed below, we have addressed all their comments and concerns, including concerns about Materials and Methods and Discussion. We have adapted the Discussion to clarify the limitations concerning the lack of representativeness of our population.
Here are my observations:
- Materials and Methods
I am sorry but the proposal: "The study design including sample size calculation is described in the study protocol and summarized in our previous paper [19]" cannot be accepted. I would ask the authors to provide this study with details about its design, where at least they should explain:
-If the participants were a convenience sample.
- The adequacy and power of the sample
- The representativeness of the sample.
Particularly in terms of representativeness, the sample seems to have a significant problem, for example, nurses are probably under-represented. Also, could the researchers support the representativeness of the sample in terms of gender, age, and type of work compared to the general PHCP population?
We thank Reviewer #3 and would like to address the points raised regarding the study design and sample characteristics. The present study is a sub-study of the national CHARMING study that was conducted with a convenience sample that included only the PHCPs participating in the CHARMING study who accepted to be recontacted for further research. In February-March 2023, the 3100 PHCPs participating in the CHARMING study who had accepted to be recontacted were invited to answer questions about their views on a periodic COVID-19 booster vaccine for themselves and their patients. A total of 1900 PHCPs (61%) responded to the survey. Given the large sample size and the high response rate, we believe the study provide meaningful insights into the views of PHCPs on periodic COVID-19 booster vaccination.
Another consideration is that the CHARMING study included GPs and the other PHCPs working in the same practice as mentioned in the following sentence in the Methods section:
“Briefly, participants eligible for this study included Belgian GPs and GP trainees practicing in primary care, as well as other PHCPs working in the same practice such as nurse and physiotherapist.” [line 89-line 91]
This is the reason why PHCPs with another type of job, e.g. nurses and physiotherapists, are under-represented relatives to the respective populations of nurses and physiotherapists in Belgium and our findings cannot be generalized to the latter populations. However, we consider PHCPs with another type of job than GP to be well represented in our sample of PHCPs working with a GP in the same practice and our findings to be generalizable to PHCPs working in a general practice.
Within each group of PHCPs distributed by type of job, we maintain the distribution of socio-demographic characteristics (age, gender, etc.) as observed in the overall study population. You can see this distribution in the attached Table (Table S1).
We have added the following statements to clarify our study population and to address its limitations:
- The Methods section of the revised version of the manuscript now included the following text on the sample:
“The present study is a sub-study of a national study (CHARMING) in which prevalence, incidence, and long-term presence of antibodies against SARS-CoV-2 both after natural infection and after vaccination were assessed among Belgian PHCPs from a convenience sample of 2001 general practices, using a rapid antibody test.” [line 85-line 88]
- The Results section and supplementary materials of the revised version of the manuscript now included the following text to clarify our study population:
“When comparing the sociodemographic information of PHCPs by type of job, we observe that there are consistently more women, workers from Flanders, and PHCPs working in big practices, regardless of the type of job (Table S1).” [line 209-line 212]
- The Discussion section of the revised version of the manuscript now included the following text on limitations related to our population:
“This study has also certain limitations, including potential selection bias because it relies on self-reported data from a convenience sample of PHCPs that accepted to be recontacted for further research. The findings may, therefore, not fully represent the PHCP population's attitudes toward vaccination.” [line 493-line 496]
“In addition, eligible participants for this study included Belgian GPs and GP trainees practicing in primary care, as well as other PHCPs working in the same practices, such as nurses and physiotherapists. However, while PHCPs with another type of job than GP are well represented in our sample of PHCPs working with a GP in the same practice, nurses and physiotherapists were underrepresented relative to their respective populations in Belgium, so our findings cannot be generalized to these groups.” [line 496-line 502]
Data and statistical analysis
In addition to the VIF, were the other assumptions for logistic regression such as independence of errors examined? Do you include the VIF value somewhere in the text?
We thank Reviewer #3. The assumptions for logistic regression were indeed examined. Multicollinearity was tested using VIF and excluded if VIF was under 4. No variables were excluded since maximum VIF factor found was 2.56. To make it clearer, the Methodology section of the revised version of the manuscript now included the following text on the VIF:
“The assumptions for logistic regression were examined with multicollinearity tested using the variance inflation factor (VIF) and excluded if VIF was under 4” [line 151-line 153]
Cook’s Distance (Di) was used to assess the influence of each observation on the overall regression model. Di<1 was considered as low influence, and no observation were removed to generate the final model. Then, to check whether the data contains potential influential observations, the standardized residual error was inspected. The data for the top 3 largest values, according to the Cook’s Distance, were the same as the one detected earlier with Di<1. Potential influential data points were then filtered with no influential data points identified. Independence of errors was respected as each participant was provided with a link to their questionnaire where they had to enter their identification code and the questionnaire could only be completed once by each participant. There are therefore no duplicate responses. To make it clearer, the Methodology section of the revised version of the manuscript now included the following text on the assumptions for logistic regression:
“The assumptions for logistic regression were examined with multicollinearity tested using the variance inflation factor (VIF) and excluded if VIF was under 4, and the influence of each observation on the overall regression model tested using Cook’s Distance (Di) and considered as low influence if Di<1. To check whether the data contains potential influential observations, the standardized residual error was inspected, and potential influential data points were then filtered. Independence of errors was respected as each participant was provided with a link to their questionnaire where they had to enter their identification code and the questionnaire could only be completed once by each participant [21].”[line 151-line 159]
- Discussion
In the discussion the researchers present their findings from an optimistic perspective, underestimating a strong anti-vaccination minority that unfortunately also exists in Belgium. However, one year after the end of the pandemic crisis, the debate on the acceptance of the COVID-19 vaccine would be outmoded if we were not concerned with this minority. Post-pandemic study identifies a link between vaccination hesitancy, anger, cynicism, and distrust in the medical system. The problem is whether behind the non-acceptance of the vaccine are hidden wrong and destructive anti-systemic views of distrust in the medical system, which will remain active even after the pandemic. Please, the authors should address this concern in the discussion.
We thank Reviewer #1 and we included this nuance regarding the minority of PHCPs that did not intend to get periodic booster vaccines for themselves in the discussion section. The Discussion section of the revised version of the manuscript now included the following text on the minority of PHCPs that did not intend to get periodic booster vaccines.
“Despite the general optimism, it is important to address the 14% of PHCPs who did not intend to get periodic booster vaccines for themselves. As introduced earlier, even though it may not be prevalent, vaccine hesitancy among HCPs exists [14]. Some studies have indicated that vaccine hesitancy is strongly associated with negative emotions such as anger and cynicism. Although they represent a minority, these groups could potentially undermine vaccine booster campaigns, even after the pandemic [31]. The ongoing prevalence of anti-establishment beliefs, encompassing conspiracy theories and a widespread lack of confidence in institutions, significantly influences vaccine acceptance. This underscores a challenge: effectively addressing vaccine hesitancy demands not only the spread of correct information but also the restoration of trust in medical establishments and tackling the deeper socio-political reasons behind this skepticism [32].” [line 429-line 439]
Add limitations of the study.
We thank Reviewer #1 and we have clarified the "strengths and limitations" section by adding a section on the strengths of our study and clarifying the limitations concerning representativeness of our population. In our study, non-GPs PHCPs, e.g. nurses and physiotherapists, are under-represented relatives to the respective populations of nurses and physiotherapists in Belgium and our findings cannot be generalized to the latter populations. However, we consider PHCPs with another type of job than GP to be well represented in our sample of PHCPs working with a GP in the same practice and our findings to be generalizable to PHCPs working in a general practice. The Discussion section of the revised version of the manuscript now included the following text on the strengths and the limitations of our study.
“The strengths of this study include that this national study is the first in Belgium to evaluate views of a large cohort of PHCPs on periodic COVID-19 booster vaccination for themselves and their patients, the reasons for acceptance or refusal, confidence in vaccine safety and efficacy as well as views on booster recommendations. In addition, a mixed method using quantitative and qualitative analysis of open-ended questions enabled this study to go further in understanding the views of PHCPs on the question of who should administer a periodic booster. Finally, this sub-study included only the PHCPs participating in the CHARMING study that accepted to be recontacted for further research. Participants were therefore familiar with the questionnaires and study team which may have limited the social desirability bias due to the nature of this study, funded by the scientific institute of public health (SCIENSANO).
This study has also certain limitations, including potential selection bias because it relies on self-reported data from a convenience sample of PHCPs that accepted to be recontacted for further research. The findings may, therefore, not fully represent the PHCP population's attitudes toward vaccination. In addition, eligible participants for this study included Belgian GPs and GP trainees practicing in primary care, as well as other PHCPs working in the same practices, such as nurses and physiotherapists. However, while PHCPs with another type of job than GP are well represented in our sample of PHCPs working with a GP in the same practice, nurses and physiotherapists were underrepresented relative to their respective populations in Belgium, so our findings cannot be generalized to these groups. The study also highlights differences in attitudes and willingness to get booster vaccines based on the region of practice (Flanders, Wallonia, Brussels). However, it does not explore the underlying reasons for these regional disparities, which could be important for public health interventions. Finally, the present study incorporates some qualitative data analysis in the section on perspectives about the authority in charge of administering booster vaccines. However, it might be beneficial to include more qualitative data analysis to gain a deeper understanding of participants' perspectives and reasons behind their attitudes.” [line 482-line 509]

Round 2
Reviewer 1 Report
Comments and Suggestions for Authors
Authors addressed partially my comments. I understand the methodological issue about variable categorizations. But I suggest to perform it as supplementary analysis in order to show the change in VH by increase of age class.
Author Response
Authors addressed partially my comments. I understand the methodological issue about variable categorizations. But I suggest to perform it as supplementary analysis in order to show the change in VH by increase of age class.
We thank Reviewer #1 for their review of our study and for providing valuable comments and suggestions. We have included an analysis of unwillingness to get a periodic COVID-19 booster vaccine based on age categories as an individual characteristic (≤30, 31-40, 41-50, 51-60, >60). This analysis is reported in Supplementary Table 2 and in the Result sections of the revised manuscript. The Results section of the revised version of the manuscript now includes the following text:
“The willingness to get a periodic COVID-19 booster did not vary with age when considered as a continuous variable. However, when age was categorized (≤ 30, 31-40, 41-50, 51-60, and > 60), participants in the 31-40 and 41-50 age groups were more likely to be unwilling to get a periodic booster compared to participants aged 30 or under (aOR 31-40 2.52, 95% CI: 1.34, 5.09 and aOR 41-50 2.46, 95% CI: 1.27, 5.08) (Table S2).” [line 280-line 285]
